# Differential Membrane Lipid Profiles and Vibrational Spectra of Three Edaphic Algae and One Cyanobacterium

**DOI:** 10.3390/ijms222011277

**Published:** 2021-10-19

**Authors:** Olimpio Montero, Marta Velasco, Jorge Miñón, Evan A. N. Marks, Aurelio Sanz-Arranz, Carlos Rad

**Affiliations:** 1Institute of Biology and Molecular Genetics (IBGM), Spanish Council for Scientific Research (CSIC), Sanz y Forés Str. 3, 47003 Valladolid, Spain; m.velasco@dicyl.csic.es; 2Composting Research Group UBUCOMP, Faculty of Sciences, University of Burgos, 09001 Burgos, Spain; jorge.minon.martinez@gmail.com (J.M.); crad@ubu.es (C.R.); 3BETA Technological Center, University of Vic-University of Central Catalonia, Edifici Can Baumann, Crta. de Roda 70, 08500 Vic, Spain; evan.marks@uvic.cat; 4Department of Fisica de la Materia Condensada, University of Valladolid, 47002 Valladolid, Spain; jausanz@gmail.com

**Keywords:** soil algae, pigments, lipids, UPLC–MS, *Klebsormidium*, *Oocystis*, *Haslea*, *Microcoleus*

## Abstract

The membrane glycerolipids of four phototrophs that were isolated from an edaphic assemblage were determined by UPLC–MS after cultivation in a laboratory growth chamber. Identification was carried out by 18S and 16S rDNA sequencing. The algal species were *Klebsormidium flaccidum* (Charophyta), *Oocystis* sp. (Chlorophyta), and *Haslea spicula* (Bacillariophyta), and the cyanobacterium was *Microcoleus vaginatus* (Cyanobacteria). The glycerolipid profile of *Oocystis* sp. was dominated by monogalactosyldiacylglycerol (MGDG) species, with MGDG(18:3/16:4) accounting for 68.6%, whereas MGDG(18:3/16:3) was the most abundant glycerolipid in *K. flaccidum* (50.1%). A ratio of digalactosyldiacylglycerol (DGDG) species to MGDG species (DGDG/MGDG) was shown to be higher in *K. flaccidum* (0.26) than in *Oocystis* sp. (0.14). This ratio increased under high light (HL) as compared to low light (LL) in all the organisms, with its highest value being shown in cyanobacterium (0.38–0.58, LL−HL). High contents of eicosapentaenoic acid (EPA, C20:5) and hexadecenoic acid were observed in the glycerolipids of *H. spicula*. Similar Fourier transform infrared (FTIR) and Raman spectra were found for *K. flaccidum* and *Oocystis* sp. Specific bands at 1629.06 and 1582.78 cm^−1^ were shown by *M. vaginatus* in the Raman spectra. Conversely, specific bands in the FTIR spectrum were observed for *H. spicula* at 1143 and 1744 cm^−1^. The results of this study point out differences in the membrane lipid composition between species, which likely reflects their different morphology and evolutionary patterns.

## 1. Introduction

Particular biochemical composition, mainly photosynthetic pigments and membrane lipids of algal species and cyanobacteria, may help to better understand the biodiversity and functioning of photosynthetic organisms. Indeed, pigments and fatty acids (FAs) have a long history of use as taxonomic biomarkers [1,2]. Nowadays, the profiling of the fatty acid composition has become relevant for biotechnological applications such as biofuels and nutritional supplements [3,4,5]. However, the thylakoid architecture and optimal photosynthetic function are dependent on the specific glycerolipid class to which molecular species the fatty acids are associated [6,7,8]. Four classes of glycerolipids make up the thylakoid membrane lipids, namely, monogalactosydiacylglycerol (MGDG), digalactosyldiacylglycerol (DGDG), sulfoquinovosyldiacylglycerol (SQDG), and diacylglycererylphosphoglycerol (PG), of which MGDG is the most abundant.

While the lipid composition of thylakoid membranes of a number of model organisms is well known, the specific composition of the photosynthetic and cell membranes of edaphic algae and cyanobacteria still remains poorly known. Fatty acids of freshwater algae follow a similar profile to that of terrestrial plants but with different proportions and marked variability [6]. Knowledge of the glycerolipid composition of phototrophs is currently based on model organisms because it allows comparison between different studies and generalizations; however, because each glycerolipid class has specific roles in the thylakoid membrane regarding photosynthesis [7,9], the diversity of photosynthetic organisms and their particular glycerolipid composition may provide insights into its essential functions and explain the different photosynthetic efficiencies, which may also be the consequence of adaptation to specific ecological niches [10]. This is to say, different mechanistic architectures may accomplish similar functions using diverse routes, even within the same algal class [11]. An important role for lipid molecules has been acknowledged with regard to the stabilization of and interaction between the protein complexes of the photosystems, with apparently specific roles for each glycerolipid class [7,12]. Thus, it is acknowledged, at present, that the membrane composition influences the dynamic physicochemical properties of the lipid bilayer, which ultimately determines the optimal photosynthetic protein organization in different organisms, according to their particular requirements [8,13]. The role played by DGDG in Light-Harvesting Complex II (LHCII) aggregation under non-photochemical quenching conditions has been shown to be relevant [14], and the role of this glycerolipid seems to be an essential requirement for the structural integrity of the oxygen-evolving complex (OEC) as well as of the photosystem I (PSI) [7]. The role played by MGDG seems to be related to its non-bilayer-forming property, which is likely involved in the LHCII assembly and coupling to the PSII, some molecules of MGDG specifically interacting with the Cyt b6/f complex [7]; additionally, MGDG solubilizes the xanthophylls involved in photoprotection through non-photochemical quenching and is necessary for xanthophyll de-epoxidase to properly effect its function [15]. There are hints regarding the fact that the abundance of MGDG and the MGDG/DGDG ratio control the photosynthetic protein organization within the operational supramolecular complexes through the “force from lipids” principle [16,17]. Both PG and SQDG confer the required negative charge to the lipid bilayer of the membrane [18]. Furthermore, three PG molecules are bound to the electron transport from plastoquinone Q_A_ to plastoquinone Q_B_, thus being related to the electron flow from PSII to PSI; additionally, SQDG has been shown to be implicated in the binding of the manganese cluster and extrinsic proteins of PSII [7]. In diatoms, SQDG is the most important glycerolipid; it controls diadinoxanthin de-epoxidation [19]. Additionally, the particular composition of fatty acyl residues of the glycerolipids has been shown to be fundamental for plant development [10,20], with the unsaturation of the fatty acyls being determinant in triggering faster phase transitions that are necessary for coping with fluctuating environmental conditions [10,21]. Recently, the regulation of the LHCII states of light-harvesting or energy dissipation as heat has been attributed to MGDG, which would switch between both states by modulating the hydrostatic lateral membrane pressure profile in the lipid bilayer [22].

At present, the implementation of mass spectrometry (MS) techniques allows for the identification of the diverse lipid species that compose the lipid profile with minimal sample handling, in particular when coupled to liquid chromatography (LC) [23,24,25,26]. Raman and infrared (IR) spectroscopies can be used to gain insights into the organism’s chemical composition, and this application has been focused over the last decade on different subjects regarding algal research [5,24,27,28,29]. These non-invasive techniques are expected to have application in diverse algae-related fields, for example, organism composition of assemblages, biotechnology, or physiology.

In this study, membrane lipid profiling by ultraperformance liquid chromatography coupled to mass spectrometry (UPLC–MS) of three algae and one cyanobacterium, grown under two illumination intensities, is reported. The four phototrophs were isolated from cropland in Burgos (Spain), where they were likely dwelling as a functional guild. Nonetheless, it should be mentioned that the organisms have been kept in laboratory cultures, either liquid or solid medium or both, for more than two years before this study was carried out. The FTIR and Raman spectra are also reported in order to gain insights into the molecular spectroscopy of diverse photosynthetic organisms.

## 2. Results

### 2.1. Species Identification

Five species of algae and three species of cyanobacteria could be distinguished in the initial filtered BG11 suspension, namely, *Klebsormidium flaccidum* (Kützing; Charophyta), *Oocystis* sp. and *Chlorella sorokiniana* (Chlorophyta), two naviculaceae (Bacillariophyceae), *Haslea spicula* and *Navicula pulchripora*, *M. vaginatus* (Cyanophyceae), and two unicellular cyanobacteria, one of them likely related to *Synechococcus* and the other to *Aphanocapsa salina*. The three algae and the cyanobacteria used in this study were chosen because there have been no or scarce data regarding their lipid profile; in addition, we were interested in comparing the lipid profiles of algae and cyanobacteria with different morphologies but dwelling under the same environmental conditions. As well, vibrational spectra were expected to show the differences in their morphology and cell organization. Photographs of the selected species for this study are shown in Figure 1. Data on similarity (%) and the sequence total score for the primers used are depicted in Appendix A.

### 2.2. Photosynthetic Pigments

The content of the representative photosynthetic pigments for the algae and cyanobacterium under the low light condition (c.a. 15 μmol photons m^−2^ s^−1^) are reported in Table 1. The relative contents of chlorophyll *b*, carotenes, and violaxanthin to chlorophyll *a* were shown to be higher in *K. flaccidum* (0.82, 0.27, and 0.15, respectively) than in *Oocystis* sp. (0.63, 0.03, and 0.10, respectively), but the opposite was shown for the ratio of lutein and neoxanthin to chlorophyll *a* (0.13 and 0.01, 0.40 and 0.12, respectively, for *K. flaccidum* and *Oocystis* sp.). In *H. spicula*, the ratio of fucoxanthin to chlorophyll *c1* was higher than that of fucoxanthin to chlorophyll *a*. The typical xanthophylls zeaxanthin and myxoxanthophyll of *M. vaginatus* were detected along with chlorophyll *a*. Representative chromatograms are depicted in Appendix A for each organism.

### 2.3. Glycerolipids

The base peak chromatogram (BPI) resulting from the UPLC–ESI–QToF–MS analysis pointed out the prevailing glycerolipids of each alga and cyanobacteria. Representative chromatograms of the lipid extract are shown in Appendix A. As well, examples of the characteristic fragmentation pattern of every glycerolipid class are illustrated in Figure 2, as measured in this study. The glycerolipid content was normalized to the chlorophyll *a* content, and it is shown in Figure 3 and Figure 4 (mol of compound/mol of chlorophyll *a*). MGDG(18:3/16:4), with *m*/*z* 789.48, was the most abundant glycerolipid in *Oocystis* sp.; it accounted for 68.6% of the MGDG species (Figure 3b and Table 2), whereas MGDG(18:3/16:3) and MGDG(18:3/16:2), with *m*/*z* 791.49 and 793.51, respectively, were predominant in *K. flaccidum* (Figure 3a) and *M. vaginatus* (Figure 4d and Table 2). Nonetheless, other MGDG and DGDG species also stood out in *M. vaginatus*. The glycerolipid profile of *H. spicula* was abundant in highly unsaturated fatty acids, mainly eicosapentaenoic acid (C20:5, EPA), with MGDG(16:4/20:5), *m*/*z* 813.48, standing out (Figure 4a). It should be mentioned that even though the fatty acid C20:5 is placed at the *sn*-2 position of the glycerol backbone for the sake of simplicity, it is known that the long-chain fatty acid is currently esterified at the *sn*-1 position in diatoms [30]. This alga exhibited SQDG(16:0/16:1) as the almost unique SQDG species; it accounted for 91.8% of this glycerolipid class (Figure 4b). Few species of PG were detected in the extracts as a whole, with C18:3, C18:2, C18:1, C16:1, and C16:0 being the fatty acyls that were esterified, and no particular PG species stood out. A few PC and PE species were also detected in the algae (Figure 3e,f and Figure 4c). Monoacylated species were only detected for the monogalactosyl class (MGMG) (Figure 3a).

Under high light (c.a. 45 μmol photons m^−2^ s^−1^), the relative content of MGDG in all the algae and the cyanobacterium was diminished as compared to low light (c.a. 15 μmol photons m^−2^ s^−1^), whereas the relative content of DGDG was enhanced, in particular in *M. vaginatus* (Table 2). SQDG content was also increased in all the algae and the cyanobacte-rium under HL compared to LL, except for *Oocystis* sp. (Table 2). The PG content rose in *K. flaccidum* and *M. vaginatus* under HL but decreased in *Oocystis* sp. and *H. spicula* (Table 2). Regarding changes in each glycerolipid species, exposure to the moderate high light level brought about a significantly (*p* < 0.01, *t*-test) substantial decrease in *K. flaccidum* of the predominant MGDG(18:3/16:3), as well as in almost all the MGDG species (Figure 3a); however, with regard to the total content of MGDG species, MGDG(18:3/16:2) and MGDG(18:2/16:2) showed a moderate, not significant, increase. Conversely, the related DGDG(18:3/16:3) and DGDG(18:3/16:2) species increased their content per chlorophyll and percentage of the glycerolipid class (Figure 3a and Table 2). In this alga, a notable decrease in the most abundant PG(18:3/16:1) was compensated for by an equivalent increase in PG(18:2/16:1) (*p* < 0.01, *t*-test) (Figure 3c and Table 2), and the anionic SQDG(18:2/16:0) also increased under HL (*p* < 0.05, *t*-test). Conversely, in *Oocystis* sp., higher light level exposure elicited increased concentrations of almost all the MGDG, DGDG, and SQDG species compounds per chlorophyll *a* content, including the standing species MGDG(18:3/16:4) (*p* < 0.05, *t*-test) (Figure 3b,d and Table 2). However, the increase in this species did not compensate for the rise in all the MGDG species and its percentage was lower under HL than under LL (Table 2). PG(18:1/16:0) and PG(18:3/16:1) also increased under higher light exposure in *Oocystis* sp.

The response of the glycerolipid species to the several light levels did not show a clear pattern in *H. spicula*; however, augmented content of MGDG(16:4/20:5) was accompanied by a concurrent decrease in contents of the remaining C20:5-containing MGDG species; an increase in the content of the C20:5-containing DGDG species seems to occur, but with statistically significant changes only being accounted for by low abundance species (Figure 4a and Table 2). A statistically significant (*p* < 0.05, *t*-test) increase in the contents of the minor species SQDG(16:1/14:1) and SQDG(16:1/16:1) was found in this alga, and the prevailing species SQDG(16:0/16:1) also rose in a per Chla basis but was not significant (*p* > 0.05, *t*-test). PG species did not vary significantly in *H. spicula*. Exposure of *M. vaginatus* to high light elicited a rise in the content of the most abundant MGDG(18:3/16:2) and DGDG(18:3/16:2) species (*p* < 0.01, *t*-test), but a concomitant decrease in the content of MGDG(18:2/16:0) was also observed. A statistically significant increase was also found for SQDG(18:3/16:0), SQDG(18:2/16:0), and PG(18:3/16:0).

The unsaturation index of the membrane (UI), this being calculated as the sum of the number of unsaturations in the two acyl chains in a given species multiplied by the content of such species, was reduced in *K. flaccidum* as a result of high light exposure (14.9 in HL versus 18.5 in LL); this reduction was likely motivated by the lower content of the MGDG species under HL than under LL (Table 3). Conversely, the other green alga, i.e., *Oocystis* sp., accounted for an increase in UI of about 45.8% under HL with respect to LL, with relevant increases of UI taking place in DGDG and anionic glycerolipid species. UI also underpinned a notable rise in *M. vaginatus* from 18.3 under LL to 24.5 under HL (33.9%), to which all the glycerolipid classes contributed. A moderate increase in UI of 8.1% took place in the diatom *H. spicula*, being mainly due to the contribution of DGDG and anionic glycerolipid species, as in *Oocystis* sp. The highest UI was counted in *H. spicu-la* and the lowest one in *Oocystis* sp. under LL, with similar values being calculated for *K. flaccidum* and *M. vaginatus*; however, under HL, *K. flaccidum* became the organism with the lowest UI.

Principal component analysis (PCA) clearly distinguished the four organisms according to their lipid profile (Figure 5a), and, in particular, *H. spicula* was separated from the other organisms by Component 1. Conversely, *K. flaccidum*, *Oocystis* sp., and *M. vaginatus* were mainly separated by Component 2. Regression parameters were R^2^X(cum) = 0.38 and Q^2^(cum) = 0.32 for Component 1, and R^2^X(cum) = 0.66 and Q^2^(cum) = 0.43 for Component 2. From the variable importance in the projection (VIP) score (Figure 5b), the significant weight of MGDG(18:3/16:4) in the *Oocystis* sp. lipid profile, and MGDG(16:4/20:5) in the *H. spicula* lipid profile were evidenced. DGDG(18:3/16:3) with *m*/*z* 953.55, DGDG(18:3/16:2) with *m*/*z* 955.56, and MGDG(18:2/16:2) or MGDG(18:3/16:1) with *m*/*z* 795.53 are shown in the VIP score to have the highest weights in *K. flaccidum* and *M. vaginatus* lipid profiles. Samples of these two latter organisms from HL and LL were distinguished though only slightly.

### 2.4. FTIR and Raman Spectroscopy

*K. flaccidum* and *Oocystis* sp. showed similar spectra between them for both Raman and FTIR spectroscopy (Appendix A), whereas *H. spicula* and especially *M. vaginatus* had characteristic peaks in the FTIR and Raman spectra, respectively. Main observable bands from the crude spectra, that is, without deconvolution and showing averaged values of frequencies from the four organisms, are illustrated in Table 4 and Table 5 for Raman and FTIR, respectively. The characteristic bands of carotenoids at ~1004, ~1157, and ~1525 cm^−1^ [27,28,31,32,33] in the Raman spectrum were detected for the four organisms. The band at the higher frequency (1521 cm^−1^), which has currently been assigned to double bond stretching, exhibited the highest intensity in the four spectra, though with comparable intensity to that of the band at 1157 cm^−1^ in *M. vaginatus*. It should be noted that a slight red shift was observed in *H. spicula* for these bands with respect to the other organisms (1013.9, 1158.5, and 1526.8 cm^−1^). Standing bands at 1629.1 and 1582.8 cm^−1^ were evidenced exclusively in the *M. vaginatus* spectrum (Appendix A) and are assigned to C=O stretching and amide I β-sheet from proteins [24,31,34], probably of the phycobilisomes (C-N stretching plus the –N-H bend of amide II [34], but these vibrations could also have a contribution from the tretrapyrrol ring of the phycobilins, which are specific components of the cyanobacteria. A weak band was shown at 1605.9 cm^−1^ in the algae, which may be indicative of the band corresponding to the vibration of the protein α-sheet [28,29,34]. Bands at 1653 cm^−1^ have been assigned to ν(C=C) stretching from unsaturated fatty acids either in pure compounds [35] or algae [29,32] and, consequently, the bands at 1654.1 cm^−1^ in *H. spicula* and 1681.1 cm^−1^ in *K. flaccidum* and *Oocystis* sp. are likely to stem from the lipid and carotenoid double bonds. A band at 1284 cm^−1^ in *M. vaginatus*, with a slight blue shift and weaker intensity in the algae (1268–1270 cm^−1^), is assigned to (C-N) stretching plus the (–N-H) bend of amide III [28,34]. Otherwise, these bands have been attributed to (=C-H) deformations in unsaturated lipids [29]. Two bands at 1327 and 1212 cm^−1^ may be assigned to chlorophyll *a* (Chla) [27,33], though these bands may arise from the overlap of several vibration modes (Table 4). Two bands were also specifically detected in *M. vaginatus* at 667 and 815 cm^−1^; these bands are currently ascribed to ring breathing from nucleotides of DNA or aromatic amino acids [28,31,36]; nonetheless, in the case of cyanobacteria, they could also arise from the tetrapyrrol ring of chlorophyll *a* or the phycobilins [24]. A weak band at 728 cm^−1^ was specifically shown in the spectrum of *H. spicula*, and this band has been ascribed to different modes of vibration, e.g., –C–S– trans [34], the tetrapyrrol ring of Chla [28], or to δ(CNH), β(NH), and δ(ring) from proteins [36]. The band at 2930 cm^−1^ has been ascribed to the C-H stretching of CH_3_ and CH_2_ groups of proteins, mainly from aromatic and aliphatic amino acids [33,37,38], but it is also representative of pectin-based carbohydrates [39]; therefore, this band could be an overlapping of C-H vibration modes coming from galactosyl groups of MGDG and DGDG plus the polysaccharides of the cell wall in addition to acyl substituents in amino acids, which would explain the band widening. All the organisms showed different intensities between HL and LL spectra in the C-H stretching band due to methyl and methylene groups at about 2930 cm^−1^, but differences in other bands between HL and LL spectra were only evident in those at 1582.8 and 1629.1 cm^−1^ in *M. vaginatus*, and in the band at 1604.0 cm^−1^ of *Oocystis* sp., whose intensity was decreased under HL (Appendix A).

FTIR spectra are shown in Appendix A, and the band assignment is depicted in Table 5. The amide I and amide II bands were detected in all the algae and the cyanobacterium at ~1638 and ~1537 cm^−1^, respectively [27,28,34]. Symmetric and asymmetric stretching (ν), as well as the CH_2_ and CH_3_ group deformations (δ) of the C-H bonds of lipids, were also evident in all the organisms. In the FTIR spectrum, specific bands were observed for *H. spicula* that correspond to the C-O and C=O vibrations from fats and carbohydrates [27,28,29,34], whereas these bands were not so evident in the remaining organisms. The algae showed relatively lower absorbance under LL than under HL, whereas the opposite happened for *M. vaginatus*, apart from the band at the highest wavenumber (c.a. 3300 cm^−1^). Decreased relative absorbance of the amide II (1537 cm^−1^) and amide I (1638 cm^−1^) bands was particularly shown in *H. spicula* and *K. flaccidum*.

## 3. Discussion

In addition to nutrient availability, other environmental factors such as UV resistance or survival mechanisms may elicit species selection or even strains within a given species [45]. Diatoms, together with green algae and cyanobacteria, are usually described as the main phototrophs in edaphic communities [45,46,47,48,49]. In this study, the lipid composition of four phototrophs from an edaphic assemblage with different architectures of the photosynthetic apparatus or morphology is reported. They include two green algae, one unicellular and one filamentous, a diatom, and a filamentous cyanobacterium.

The two green algae showed a similar lipid profile between them, but specific variations in species within each lipid class suggest a somewhat different organization of the photosynthetic apparatus. These subtle differences may account for variations in photosynthetic efficiency and capability to respond to inhibitory conditions (e.g., xanthophyll cycle and non-photochemical quenching). Indeed, principal component analysis (PCA) placed each alga separately (Figure 5). The photosynthetic apparatus of *Oocystis* sp. seems to be highly dependent on DGDG species, which had higher diversity than in *K. flaccidum*, with the highly unsaturated MGDG(18:3/16:4) being the major building block of the photosynthetic membranes and a substantial decrease of the MGDG/DGDG ratio under HL (4.41) compared to LL (7.14). In *K. flaccidum*, two slightly less unsaturated MGDG species were abundant, namely, MGDG(18:3/16:3) and MGDG(18:3/16:2), with the content of the less unsaturated species increasing under HL. In this latter alga, the ratio of MGDG to DGDG also dropped under HL (3.82) compared to LL (2.87), but to a lesser extent than in *Oocystis* sp. Given that DGDG is tightly associated with the LHCII complex and, in particular, with the water-splitting complex [7,15,50], the rise in the content of DGDG species under HL suggests changes in the organization of the photosynthetic apparatus regarding higher LHCII aggregation and hydrophobic mismatch [14]. However, this result is somewhat contradictory with the expected involvement of non-bilayer lipids, i.e., MGDG, in the membrane energization state [15,22]. The high decrease in the MGDG/DGDG ratio of *Oocystis* sp. under HL compared to LL may be related to a different dominant lipid phase, with an inverted hexagonal (H_II_) form under LL but increasing the bilayer surface under HL, which is noteworthy given that the HL condition used in this study may not be considered stressing for photosynthetic performance. Conversely, in *K. flaccidum*, the increase in the lipid bilayer surface was not so pronounced, a fact that may suggest that the compensation point of photosynthesis (E_k_ in the P-I curve) is attained at a higher irradiance in *K. flaccidum* than in *Oocystis* sp. Furthermore, the reduction in the unsaturation index of *K. flaccidum* under HL compared to LL is surprising, which might be related to a more stable organization of the thylakoids that facilitates membrane fluidity. *K. flaccidum* is a filamentous alga, whereas *Oocystis* sp. is unicellular (Figure 1); thus, it may be hypothesized that the chloroplasts of the cells of a given filament are interconnected and respond as a whole, which implies that the measured composition is averaged between more exposed cells to light and the cells undergoing self-shading, whereas all *Oocystis* sp. cells are exposed and self-dependent, which, consequently, likely renders a more homogeneous response to the different light intensities. These results point out that within the general architecture of the photosynthetic apparatus, subtle differences may exist between organisms, which are likely related to their specific physiological and morphological characteristics and ultimately become determined by the particular requirements of each organism for adaptation to the conditions of the ecosystem where they currently dwell.

In *Oocystis* sp., the relative content (about 1% of total glycerolipids) of PG species did not change from LL to HL conditions. However, the most abundant PG under LL, namely, PG(18:3/16:0), underwent a drastic reduction of its content under HL (from 33.89% to 23.55% of total PGs), whereas the opposite happened with the less unsaturated species PG(18:1/16:0), whose content rose from 23.57% to 42.93%. A similar trend was observed in *K. flaccidum* but with the PG species PG(18:3/16:1) and PG(18:2/16:1) instead. Since PG has been related to the functional integrity of PSI and to the PSI/PSII coupling [51,52], these changes might be due to a reorganization of the electron transfer between both photosystems. It is noteworthy that fewer PG species were detected in *K. flaccidum* than in *Oocystis* sp., and C16:0 was dominant in this latter alga at the *sn-2* position of the glycerol backbone whilst it was C16:1 in *K. flaccidum*. This differential composition of C16 fatty acyls between both algae suggests again differences in the thylakoid membrane structure and organization that might be related to morphological and eco-physiological adaptation. A somewhat similar pattern to that of PG species was observed in SQDG species in *Oocystis* sp., but no relevant changes were observed in *K. flaccidum* for the SQDG species between LL and HL. The rise in less saturated acyl chains, with the concomitant drop of more unsaturated acyl chains, in the negatively charged glycerolipids (that is, PG and SQDG) is likely to be related to the structural organization of the PSII complex and the assembly of the extrinsic proteins in the oxygen-evolving complex as well as in the electron transport between the plastoquinones Q_A_ and Q_B_ [7,53].

The glycerolipid profile of *M. vaginatus* resembled that of cyanobacteria [24,54,55,56], thus suggesting a similar photosynthetic membrane architecture. However, more DGDG species were detected in *M. vaginatus* in the present study than those previously reported for a marine strain of *Synechococcus* sp. [24], which may suggest a higher dependence on plasticity for the required changes of the oxygen-evolving complex (OEC) and LHCII aggregation to respond to environmental changes [53]. The filamentous character of *M. vaginatus* may also be a reason for that. The same acyl changes were shown in the most abundant species of both MGDG and DGDG, a fact that could be related to a capacity for fast acclimation to changes in environmental conditions associated with variations in the membrane phase. The conversion of MGDG into DGDG relies only on the addition of an additional galactosyl group, but this fact induces a relevant physicochemical change in the membrane, which is the conversion of non-bilayer-forming MGDG to bilayer-forming DGDG and, consequently, a lipid phase transformation [10,15]. However, the relative proportion of these most abundant DGDG species did not change with the light intensity. The MGDG/DGDG ratio was 2.64 and 2.15 under LL and HL, respectively; provided that this ratio is independent of the chlorophyll content or cell number, it seems evident that high light conditions imply a higher DGDG content, a fact that is likely related to an increase in the PSI/PSII ratio and enhanced thermotolerance [7,57]. Swelling of the thylakoid lumen under illumination in cyanobacteria has been reported [58], and this feature implies a reduction of the membrane curvature, a fact that is likely to be associated with the increase in the content of the bilayer-forming DGDG species.

As is well known, the composition and structure of the thylakoid membrane of diatoms differ substantially from those of green and other algal classes [12,19,59,60,61]. In diatoms, fucoxanthin plays a more active role than that played by the xanthophylls in the green algae [11], with fucoxanthin being the major contributor to the fucoxanthin protein complexes (FPC), along with chlorophyll *a*. The high ratio of fucoxanthin:Chla, measured spectrophotometrically in this study (0.60) under quite a low irradiance (see Table 1), is in agreement with such a notion. Typical features regarding the membrane lipid composition of diatoms were also shown for *H. spicula* in this study. They include dominance of MGDG and DGDG species with long chain highly unsaturated fatty acyls, mainly C20:5 (EPA) and C20:4 (AA), a near complete absence of C18 fatty acyls, and abundant C16 fatty acyl chains with up to four double bonds in addition to the prevalence of SQDG with C16 acyl chains instead of PG [10,19]. Indeed, SQDG(16:0/16:1) accounted for 91.8% and 73.9% of SQDGs and 78.8% and 89.1% of SQDG + PG, under LL and HL, respectively. As a general pattern, SQDG species rose, whereas PG species remained unchanged under HL compared with LL, which is in agreement with the relevant role played by SQDG in diatoms regarding the inhibition of diadinoxanthin de-epoxidation as opposed to the shield of MGDG surrounding the FPCs that favors diadinoxanthin and diatoxanthin solubilization [19,62]. Nonetheless, MGDG and DGDG species with a C18/C16 composition were reported to be abundant in *Haslea ostrearia*, whereas C20:5/C16:n species were a minority [30].

The Raman spectrum is more stable and sensitive to non-polar compounds, whereas FTIR spectrum is more reliable and sensitive to polar compounds (i.e., proteins) [27]. Therefore, major signals in FTIR are due to proteins, which account for about 70% of the thylakoid membrane area [8], and saccharides, whereas major bands in Raman are due to lipids and especially to double bonds from carotenoids and fatty acyl chains of glycerolipids (Table 4 and Table 5). Nonetheless, aliphatic chains of amino acids may contribute to Raman signals from proteins as well [33]. The subtle differences in lipid composition and, possibly, thylakoid membrane organization between *K. flaccidum* and *Oocystis* sp. were not reflected in the vibrational spectra. Thus, both algae showed overlying FTIR and Raman spectra. Nonetheless, two features could help distinguish between the algae, given that they have a differential Raman relative intensity in the high-frequency range around 2930 cm^−1^, which could be related to a high content of galactolipids in *Oocystis* sp. compared to *K. flaccidum* and a ratio of the band at 1638 cm^−1^ to the band at 1040 or 1045 cm^−1^ in the FTIR spectrum, which is higher than 1 for *Oocystis* sp. and lower than 1 for *K. flaccidum* (Appendix A), though this feature might depend on the nutritional status [29,43,63].

The major Raman bands at 1157 and 1524.9 cm^−1^, observed in the two green algae, were also shown in *H. spicula* and *M. vaginatus* (Table 4), a fact that shows the relevance of carotenoids in Raman spectra. However, the Raman spectrum of *M. vaginatus* exhibited four specific bands, which may likely arise from the phycobilisomes (PBSs), a distinctive feature of cyanobacteria. Indeed, the reduction of the relative intensity of the bands at 1582.8 and 1629.1 cm^−1^ under HL compared to LL supports such a conclusion. The typically reported FTIR bands of cyanobacteria were also shown for *M. vaginatus* in this study [42,44]. Conversely, specific bands could be ascribed to *H. spicula* in the FTIR spectrum, which can be derived from the fucoxanthin–protein complexes (FPCs), being, in turn, specific features of diatoms [11,19,62,64]. Using resonance Raman spectroscopy with different excitation wavelengths, the organization of FPCs in the diatom *Cyclotella meneghiniana* has been studied with regard to both the carotenoid binding and oligomerization of the FPCs [11,65]. These studies have shown fucoxanthin molecules specifically binding to the FPCa or the FPCb complexes, with fucoxanthin molecules in three different excitation stages that mainly absorb in the blue, red, or green wavelengths. The prevalence of diadinoxanthin binding to FPCb could also be discerned [65]. Two ring-breathing modes of chlorophyll c2 were also shown to rely on the trimeric or the oligomeric forms of the FPCs. The different organization of the thylakoids between green algae and diatoms [62] gives rise to specific bands for the same chemical groups, depending on the particular environment, as this fact may shift the absorption maxima. In this regard, specific bands were observed for *H. spicula* in the FTIR spectrum, which may be due to chlorophyll *c* and fucoxanthin, two molecules that are not present in the green algae and the cyanobacteria.

## 4. Materials and Methods

### 4.1. Experimental Setup

Algae were collected from an agricultural field in Burgos (Spain) with geographical coordinates 42°29′44.2″ N (latitude) and 3°48′17.5″ W (longitude). The soil is classified as calcaric cambisol (CMc) according to the FAO [66]. The main chemical properties are: pH (water 1:2.5 *w*/*v*) 7.9; electrical conductivity (water 1:5 *w*/*v*, 25 °C) 0.498 dS m^−1^; soil organic matter 3.92%; total nitrogen 2.28 g kg^−1^. For algae collection, a small piece of geotextile polypropylene fiber (120 g m^−2^) was placed in a run-off zone of the cultivated field and left for three weeks. After this period, the textile fiber was removed, immersed in BG-11 culture medium, and transported to the laboratory for isolation of the algae and microscope inspection. Five aliquots of the liquid suspension were obtained from the fiber-derived culture and inoculated in agar plates with BG-11 medium. Subsequent sub-samplings of diverse colonies in the agar cultivations obtained in the initial agar plate cultures were carried out until non-axenic isolates of each algal species were obtained, visually confirmed by microscopic inspection.

The diverse algal species were identified by morphological inspection using a micro-scope and 16S or 18S rDNA sequencing, along with the phylogenetic tree position for the forward + reverse primer similarity, and the determination of the photosynthetic pigment profile by means of high-pressure liquid chromatography with diode-array detection (HPLC–DAD,Thermo Scientific, Madrid, Spain).

Samples used in this study for biochemical analysis were obtained from BG-11 liquid medium cultures for all the algae and from BG-11 solid agar medium in the case of *M. vaginatus*. All cultures of algae and cyanobacteria were kept in a culture chamber with an illumination of ~15 (low light, LL condition) or ~45 μmol photon m^−2^ s^−1^ (high light, HL condition), and 20 °C. The algal and cyanobacterial cells were allowed to grow for at least three growth cycles under each illumination before they were used for the experiments.

### 4.2. Photosynthetic Pigment Measurement

Photosynthetic pigments in a methanolic extract were analyzed by high-performance liquid chromatography with photodiode array detection (HPLC–DAD) using the same chromatographic method as in Montero et al. [24]. A FINNIGAN SURVEYOR PLUS chromatography system (Thermo Scientific) equipped with a Quaternary LC pump, autosampler, and PDA detector was used for the HPLC–DAD measurements. Pigments were identified according to the retention time and UV–Vis spectrum (350–700 nm). Chlorophyll *a* (Chla), zeaxanthin (Z), and β-carotene (bC) were quantified after regression curves were drawn using commercial standards from SIGMA-ALDRICH, Merck KGaA, Darmstadt, Germany (references are C5753 for Chla, 1733122 (USP) for Z, and 1065480 (USP) for bC). Other chlorophylls were also quantified using the Chla regression curve. The regression curve of Z was used for the quantification of other xanthophylls as well. Each pigment was quantified at its own maximum absorption wavelength (Max-Plot).

### 4.3. UPLC-QToF-MS Measurements

A dichloromethane:methanol (2:1, *v*/*v*) extract was analyzed by ultra-performance liquid chromatography coupled to quadrupole time-of-flight mass spectrometry (UPLC–QToF–MS), as in Montero et al. [24]. After lipid extraction, the supernatant was evaporated to dryness and the pellet resuspended in methanol:water (9:1, *v*/*v*). An Acquity™ UPLC system (WATERS, Manchester, UK) equipped with an automatic injector (Sample Manager) and a binary solvent pump (Binary Solvent Manager) was used for liquid chromatography. The output of the liquid chromatographer was connected to a SYNAPT G2 HDMS mass spectrometer (WATERS, Manchester, UK), with a time-of-flight analyzer (QToF) and an electrospray ionization source (ESI). The chromatographic column was an Acquity UPLC BEH HSS T3 100 × 2.1 mm, 1.7 μm p.s., with a 10 × 2.1 mm precolumn (vanguard column) of the same stationary phase. Solvents were (A) methanol:water:formic acid (50:50:0.5, *v*/*v*/*v*) and (B) methanol:acetonitrile:formic acid (59:40:0.5, *v*/*v*/*v*), both with 5 mM ammonium formate. Compounds were eluted at a flow rate of 0.35 mL/min using the gradient that follows: initial, 100% A; 1 min, 100% A; 2.5 min, 20% A; 4 min, 20% A; 5.5 min, 0.1% A; 8.0 min, 0.1% A; 10 min, 100% A; this was kept isocratic for 2 min to recover initial pressure before the next injection. Samples were analyzed with negative ionization using an MS^E^ method [67]. Compounds were identified by the *m*/*z* value, the elemental composition compatible with the isotopic distribution and relative retention time, and specific fragments from the MS^E^ function. Samples from three independent cultures were analyzed for each alga and the cyanobacterium. For glycerolipid quantification, the commercial standards that follow were used: from Avanti Polar Lipids Inc (Alabaster, Alabama, EE.UU.)., monogalactosyldiacylglycerol, with an averaged molecular weight of 775.06 (CAS No. 1932659-76-1 and reference SKU 840523P-5 mg), and sulfoquinovosyldiacylglycerol, with a molecular weight of 834.152 (CAS No. 123036-44-2 and reference SKU 840525P-5 mg); from Larodan Research Grade Lipids (Solna, Sweden), 1,2-dimyristoyl-*sn*-glycero-3-phosphatidylglycerol Na salt (CAS No. 200880-40-6 and reference 38-3014-9), 1,2-dilauroyl-*sn*-glycero-3-Phosphatidylcholine (CAS No. 18656-40-1 and reference 37-1200-9) and 1,2-dilauroyl-*sn*-glycero-3-phosphatidylethanolamine (CAS No. 59752-57-7 and reference 37-1220-9). Fucoxanthin was quantified using the zeaxanthin standard, and chlorophylls were quantified using the Chla standard from SIGMA-ALDRICH, indicated above.

### 4.4. Fourier Transform Raman (FT-Raman) and ATR-Infrared (ATR-FT-IR) Spectroscopies

Attenuated total reflectance Fourier transform infrared (ATR–FT–IR) and FT–Raman spectra were done from both liquid medium and solid medium cultures for the algae. Aliquots of 5 mL volume of the algal suspension were filtered through PTF membranes, and the cell biomass was immediately placed on special devices used for the spectroscopic measurements of FT–Raman and ATR–FT–IR. For the solid medium cultures, an amount of *M. vaginatus* biomass or algae biomass was taken from the agar plates using a disposable microstreaker and placed on the special device for the spectroscopic measurements. Samples from two independent cultures were analyzed for every illumination. A Perkin Elmer Spectrum 100 FT-IR spectrometer with a Universal ATR sampling accessory was used for the ATR–FT–IR measurements. Spectral resolution was 4 cm^−1^, and 16 cumulative spectra were acquired for each measurement. FT–Raman spectra were acquired with an FT–Raman Bruker RFS100/S instrument equipped with a LASER Klastech, Senza series (1064 nm, 500 mW), and a CCD Bruker D418-T (range 851–1695 nm),(Bruker Española S.A., Rivasvaciamadrid, Madrid, Spain).. The diameter of the LASER spot over the sample was 1000 μm. Other parameters were 10 mm slit, a scan rate of 1.6 kHz, and number of accumulations of 1024 [24]. All samples were harvested just a few minutes before the measurement was carried out.

### 4.5. Statistical Analysis

Pairwise comparisons of the diverse lipid contents between low (LL) and high (HL) light were done using the Student *t*-test, and significant differences were accepted for *p* < 0.05. Samples from three independent cultures (*n* = 3) were measured for each illumination.

Principal component analysis (PCA) of the UPLC–QToF data was done using the Extended Statistic (XS) application included in MarkerLynx^®^ software (V4.1 SCN 803, WATERS, Manchester, UK). This application includes the statistical tools of the SIMCA-P+ software package (Umetrics EZ info 2.0; Umea, Sweden). The statistical parameters R^2^X(cum) and Q^2^(cum), which explain the variability of X-variables and indicate model predictive capability, respectively, were determined [68]. Previously, a three-dimensional data array (Pareto-scaled) comprising the variables sample (including the blanks), retention time_*m*/*z* values (molecular features), and normalized (scaled to Pareto variance) signal intensity of the *m*/*z* value was generated using MarkerLynx^®^ software (WATERS, Manchester, UK).

## 5. Conclusions

The membrane lipid profile of two green algae, one filamentous and one unicellular, a diatom, and a cyanobacterium from an edaphic assemblage, is shown. They differ in the relative proportion of the main glycerolipid classes and the response to low and high light exposure. The differential lipid composition between them allowed their clear separation in principal component analysis (PCA). The high abundance of MGDG(18:3/16:4) and the high variety of DGDG species with a high DGDG/MGDG ratio in *Oocystis* sp. compared to *K. flaccidum* suggest subtle differences in the architecture of the photosynthetic apparatus between the two green algae. Nonetheless, these subtle differences in lipid composition were not clearly reflected in the Raman and FTIR spectra. The typical lipid composition of diatoms, with an abundant content of C20:5 and C20:4 acyls in MGDG and DGDG species, was also found for *H. spicula* in this study. This alga showed specific bands in the FTIR spectrum, whereas *M. vaginatus* showed specific bands in the Raman spectrum.

## Figures and Tables

**Figure 1 ijms-22-11277-f001:**
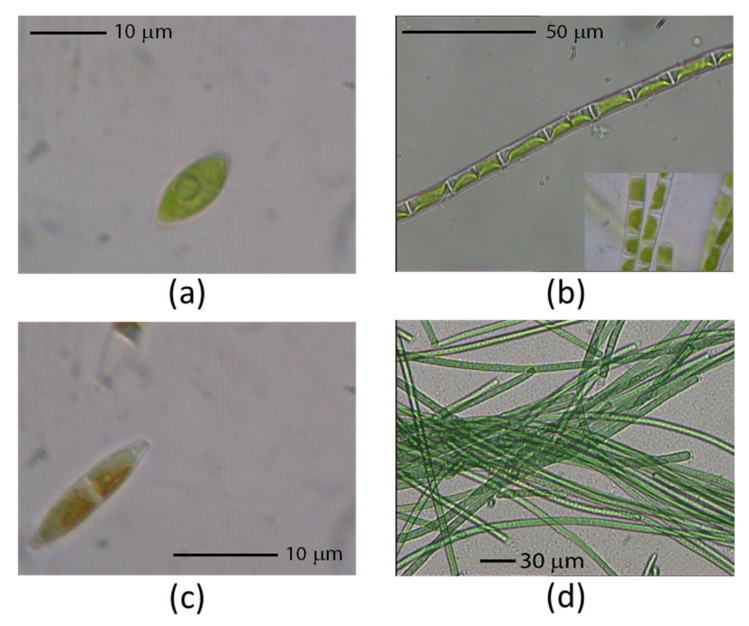
Photographs of the algal species used in this study. (**a**) *Oocystis* sp. (cell length c.a. 14.7 μm); (**b**) *Klebsormidium flaccidum* (cell length c.a. 16.9 μm); (**c**) *Haslea spicula* (cell length c.a. 15.8 μm); (**d**) *Microcoleus vaginatus* (cell length undetermined). A Leica MDLB microscope with a LEICA MC 170 HD camera and image capture software (IM50 1.20R19) were used for photographs (40×).

**Figure 2 ijms-22-11277-f002:**
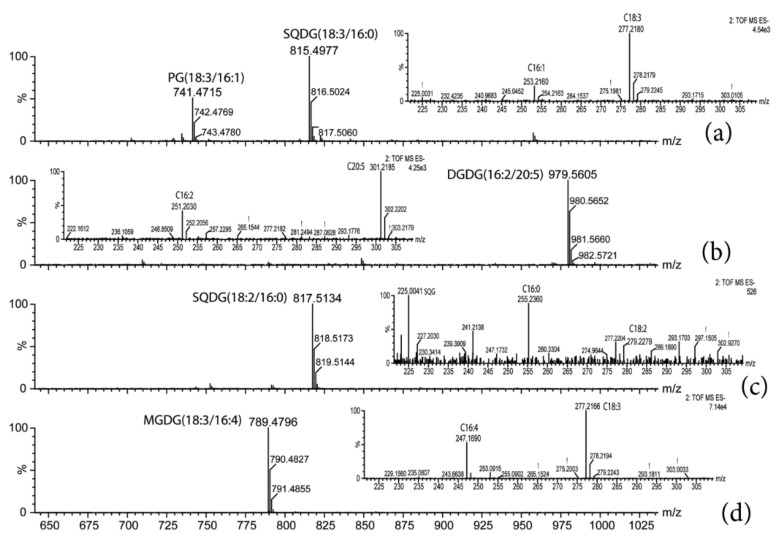
Mass spectra obtained in low energy function (full-scan) and in high energy function 2 (MSE; figure insert) of some relevant glycerolipid species. The [M-H]- ions of the fatty acyls esterifying the corresponding glycerolipid species shown in the full-scan are pointed out in the inserted figure. (**a**) diacylglycerylphosphoglycerol (PG), (**b**) digalactosyldiacylglycerol (DGDG), (**c**) sulfoquinovosyldiacylglycerol (SQDG), and (**d**) monogalactosyldiacylglycerol (MGDG).

**Figure 3 ijms-22-11277-f003:**
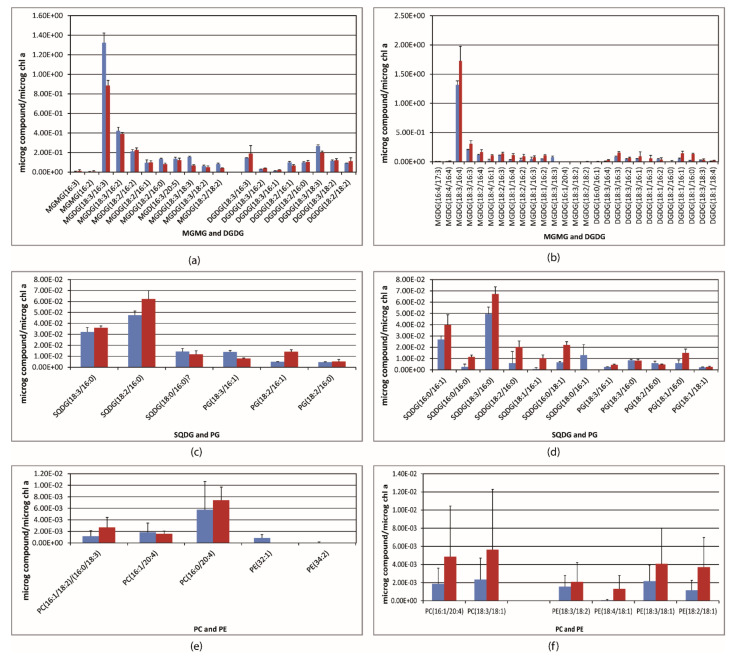
Content of the different glycerolipids detected in the extract of the algae *K. flaccidum* (panels **a**,**c**,**e**) and *Oocystis* sp. (panels **b**,**d**,**f**). MGMG: monogalactosylmonoacylglycerol; MGDG: monogalactosyldiacylglycerol; DGDG: digalactosyldiacylglycerol; SQDG: sulfoquinovosyldiacylglycerol; PG: diacylglycerylphosphoglycerol; PC: diacylglycerylphosphocholine; PE: diacylglycerylphosphoethanolamine. Values are the mean ± standard deviation of three independent cultures (*n* = 3). Blue bars, low light; red bars, high light. See also Annex A in the Appendix A.

**Figure 4 ijms-22-11277-f004:**
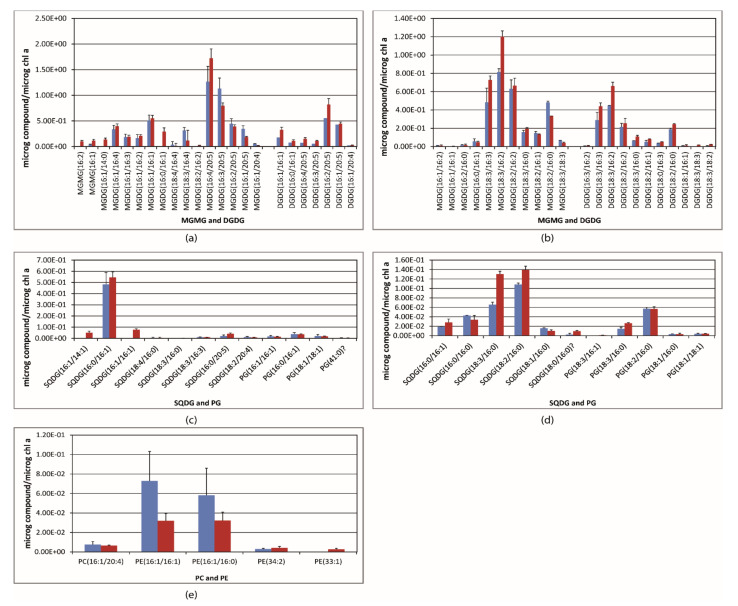
Content of the different glycerolipids detected in the extract of the algae *H. spicula* (panels **a**,**b**,**c**) and *M. vaginatus* (panels **d** and **e**). MGMG: monogalactosyl-monoacylglycerol; MGDG: monogalactosyldiacylglycerol; DGDG: digalactosyldiacylglycerol; SQDG: sulfoquinovosyldiacylglycerol; PG: diacylglycererylphosphoglycerol; PC: diacylglycererylphosphocholine; PE: diacylglycererylphosphoethanolamine. Values are the mean ± standard deviation of three independent cultures (*n* = 3). Blue bars, low light; red bars, high light. See also Annex A in the Appendix A.

**Figure 5 ijms-22-11277-f005:**
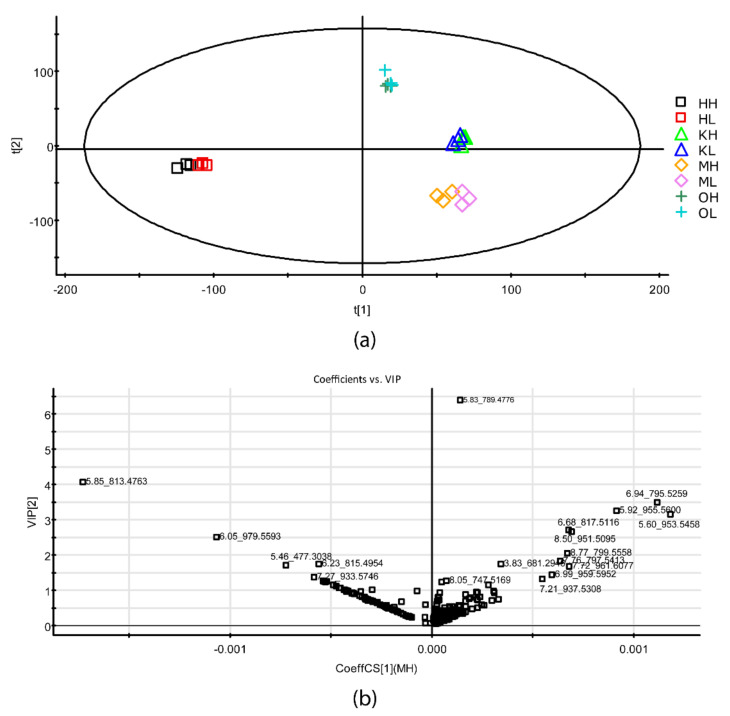
(**a**) Score-plot obtained in the principal component analysis (PCA) of the four organisms’ lipid profiles. (**b**) Variable importance in projection (VIP) score of the different lipid species. Legend of Panel (**a**): first character: H, *Haslea spicula*; K, *Klebsormidium flaccidum*; O, *Oocystis* sp.; M, *Microcoleus vaginatus*; second character: H, high light and L, low light.

**Table 1 ijms-22-11277-t001:** Contents of representative photosynthetic pigments in each alga and cyanobacteria as determined by means of HPLC–DAD measurements. Abbreviations: Kfl, *Klebsormidium flaccidum*; Hsp, *Haslea spicula*; Ooc, *Oocystis* sp.; Mic, *M. vaginatus*; Hex, hexanoyl; and Oct, octanoyl.

Pigment	ng Pigment/mL Culture			
	Kfl	Ooc	Hsp	Mic
Neoxanthin	96.01	303.40		
Violaxanthin	1041.76	256.39		
Lutein	894.40	1051.57		
Zeaxanthin				90.86
Diadinoxanthin			7.75	
Diatoxanthin			5.84	
Fucoxanthinol			19.70	
Fucoxanthin			81.21	
Hex-fucoxanthin			2.86	
Oct-fucoxanthin			3.16	
Myxoxanthophyll				276.99
Chlc1			7.94	
Chlb	5755.25	1630.24		
Chla	7015.92	2601.65	135.94	4760.75
Carotenes	1918.27	67.72	22.24	485.70

**Table 2 ijms-22-11277-t002:** Glycerolipid species detected in the algal and cyanobacterial extracts. The averaged percentages of a given glycerolipid species within the corresponding glycerolipid class and of a given class with regard to the total glycerolipid (∑GLs = MGDG + DGDG + SQDG + PG) content are provided (*n* = 3). Upper and lower values correspond to the low and high light conditions, respectively. Nomenclature: MGDG, monogalactosyldiacylglycerol; DGDG, digalactosyldiacylglycerol; SQDG, sulfoquinovosyldiacylglycerol; PG, diacylglycerylphosphoglycerol; PC, diacylglycerylphosphocholine; PE, diacylglycerylphosphoethanolamine. Abbreviations: Kfl, *Klebsormidium flaccidum*; Hsp, *Haslea spicula*; Ooc, *Oocystis* sp.; Mic, *Microcoleus vaginatus*. The *p*-value of the Student *t*-test for the comparison between high light and low light data is indicated as a superscript for values <0.05. Species marked with the symbol § superscript could not be accurately identified. N.Q. = not quantifiable.

*m/z*	Elemental Composition	Compound	Each Species Content/Sum of the Class Species Content (Percentage)			
			Kfl	Hsp	Ooc	Mic
**MGDG**						
531.2801	C25H42O9	MGMG(16:3/0:0)	0.19 ± 0.33 0.92 ± 0.94	-	-	-
533.2927	C25H44O9	MGMG(16:2/0:0)	0.13 ± 0.22 0.54 ± 0.60	0.00 ± 0.00 ^<0.001^ 2.84 ± 0.53	-	-
535.3118	C25H46O9	MGMG(16:1/0:0)	-	1.15 ± 0.28 ^0.006^ 3.24 ± 0.81	-	-
745.5102	C39H72O10	MGDG(16:1/14:0)	-	0.00 ± 0.00 ^0.001^ 2.72 ± 0.56	-	-
765.4796	C41H68O10	MGDG(16:1/16:4)	-	6.72 ± 1.66 7.64 ± 0.87	-	-
767.4964	C41H70O10	MGDG(16:1/16:3)/(16:2/16:2)	-	3.75 ± 1.14 3.72 ± 0.53	-	-
769.5097	C41H72O10	MGDG(16:1/16:2)	-	3.24 ± 1.46 4.04 ± 0.42	-	0.42 ± 0.07 0.32 ± 0.28
771.5266	C41H74O10	MGDG(16:1/16:1)	-	10.18 ± 2.13 10.53 ± 1.12	-	0.00 ± 0.00 0.05 ± 0.09
771.5278	C41H74O10	MGDG(16:2/16:0)	-	-	-	0.62 ± 0.25 0.50 ± 0.32
773.5392	C41H76O10	MGDG(16:0/16:1)	-	6.07 ± 5.76 5.60 ± 1.36	-	2.03 ± 1.00 1.50 ± 0.32
775.4636	C42H66O10	MGDG(16:4/17:3)	-	-	0.27 ± 0.12 0.16 ± 0.10	-
787.4642	C43H66O10	MGDG(18:4/16:4)	-	1.19 ± 0.27 0.94 ± 0.16	0.17 ± 0.070 ^0.006^ 0.38 ± 0.09	-
789.4804	C43H68O10	MGDG(14:3/20:4)	-	N.Q.	-	-
789.4801	C43H68O10	MGDG(18:3/16:4)	-	6.12 ± 1.15 2.17 ± 3.76	68.62 ± 3.60^0.027^ 59.11 ± 8.64	-
791.4965	C43H70O10	MGDG(18:3/16:3)	50.09 ± 3.76 ^0.001^ 44.70 ± 2.69	-	10.51 ± 0.52 ^0.015^ 10.41 ± 1.82	17.01 ± 5.39 ^0.029^ 21.65 ± 1.32
791.4971	C43H70O10	MGDG(18:2/16:4)	-	-	6.01 ± 0.63 ^0.041^ 5.74 ± 1.28	-
793.5113	C43H72O10	MGDG(18:4/16:1)	-	N.Q.	1.13 ± 1.13 ^0.002^ 3.75 ± 0.40	-
793.5113	C43H72O10	MGDG(18:3/16:2)	15.94 ± 1.42 19.66 ± 0.73	-	-	28.47 ± 1.31 ^<0.001^ 35.63 ± 1.85
793.5115	C43H72O10	MGDG(18:2/16:3)	-	-	5.64 ± 0.32 ^0.042^ 4.74 ± 0.75	N.Q.
793.5102	C43H72O10	MGDG(18:1/16:4)	-	-	1.41 ± 0.62 ^0.003^ 3.85 ± 0.90	-
795.5261	C43H74O10	MGDG(18:2/16:2)	7.97 ± 0.73 11.38 ± 1.07	0.27 ± 0.27 0.40 ± 0.13	1.61 ± 1.47 ^0.043^ 3.14 ± 1.27	22.00 ± 3.44 19.67 ± 2.44
795.5283	C43H74O10	MGDG(18:3/16:1)	-	-	N.Q.	-
795.5283	C43H74O10	MGDG(18:1/16:3)	-	-	2.27 ± 2.10 2.73 ± 0.80	-
797.5432	C43H76O10	MGDG(18:3/16:0)	1.13 ± 0.98 0.00 ± 0.00	-	-	5.47 ± 0.73 ^0.029^ 5.74 ± 0.38
797.5432	C43H76O10	MGDG(18:2/16:1)	3.64 ± 1.09 4.97 ± 0.86	-	-	5.12 ± 0.67 3.93 ± 0.17
797.5423	C43H76O10	MGDG(18:1/16:2)	-	-	2.28 ± 0.36 ^0.001^ 3.71 ± 0.39	-
799.5585	C43H78O10	MGDG(18:2/16:0)	5.00 ± 0.33 ^0.001^ 4.07 ± 0.52	-	0.00 ± 0.00 1.91 ± 1.76	16.71 ± 0.54 ^<0.001^ 9.84 ± 0.05
813.4788	C45H68O10	MGDG(16:4/20:5)	-	23.95 ± 5.61 ^0.044^ 31.07 ± 3.33	-	-
815.499	C45H70O10	MGDG(16:3/20:5)/(16:4/20:4)	4.98 ± 0.60 5.97 ± 1.02	21.31 ± 3.88 ^0.027^ 14.33 ± 0.99	-	-
817.5106	C45H72O10	MGDG(16:2/20:5)	-	8.44 ± 1.80 7.04 ± 0.63	-	-
817.5094	C45H72O10	MGDG(18:4/18:3)	-	-	N.Q.	-
819.5266	C45H74O10	MGDG(16:1/20:5)	-	6.52 ± 1.06 ^0.004^ 3.36 ± 0.21	-	-
819.5266	C45H74O10	MGDG(18:3/18:3)	5.65 ± 0.31 ^<0.001^ 3.32 ± 0.35	-	N.Q.	2.24 ± 0.13 ^0.004^ 1.22 ± 0.21
821.5436	C45H76O10	MGDG(16:1/20:4)	-	1.04 ± 0.07 ^<0.001^ 0.34 ± 0.05	-	-
821.5436	C45H76O10	MGDG(18:3/18:2)	2.19 ± 0.43 2.40 ± 0.67	-	-	+
823.5602	C45H78O10	MGDG(18:2/18:2)	3.01 ± 0.37 ^0.001^ 1.91 ± 0.22	-	0.00 ± 0.00 ^0.001^ 0.26 ± 0.07	-
		Sum MGDG/∑GLs	76.71 ± 13.21 ^0.002^ 70.73 ± 9.65	75.06 ± 19.94 69.31 ± 10.72	82.97 ± 9.11 ^0.008^ 77.33 ± 14.15	67.33 ± 9.09 ^0.004^ 63.03 ± 4.70
**DGDG**						
721.3647	C33H56O14	DGMG(18:3/0:0)	-	-	-	-
927.5306	C47H78O15	DGDG(16:3/16:2)	-	-	-	0.36 ± 0.31 0.53 ± 0.13
933.5787	C47H84O15	DGDG(16:1/16:1)	-	13.22 ± 4.82 ^0.015^ 17.20 ± 2.42	-	-
935.5947	C47H86O15	DGDG(16:0/16:1)	-	5.74 ± 2.31 5.90 ± 0.93	0.00 ± 0.00 ^0.020^ 0.49 ± 0.29	N.Q.
951.5333	C49H78O15	DGDG(18:3/16:4)	-	N.Q.	2.80 ± 0.86 ^0.001^ 3.74 ± 0.57	-
953.5494	C49H80O15	DGDG(18:3/16:3)	17.30 ± 0.23 22.35 ± 10.06	-	25.52 ± 4.20^0.002^ 19.61 ± 2.10	22.01 ± 6.66 ^0.026^ 23.13 ± 2.03
955.5649	C49H82O15	DGDG(18:3/16:2)	3.62 ± 0.12 ^<0.001^ 4.82 ± 0.12	-	13.01 ± 3.44 7.90 ± 1.70	34.07 ± 0.13 ^<0.001^ 34.72 ± 2.19
957.5807	C49H84O15	DGDG(18:3/16:1)	1.39 ± 0.31 ^0.003^ 2.72 ± 0.31	-	14.95 ± 1.94 11.06 ± 9.97	-
957.5788	C49H84O15	DGDG(18:2/16:2)	-	-	-	N.Q.
957.5770	C49H84O15	DGDG(18:1/16:3)	-	-	0.00 ± 0.00 6.95 ± 6.53	-
959.5973	C49H86O15	DGDG(18:3/16:0)	-	-	-	4.85 ± 0.16 ^0.002^ 5.73 ± 0.69
959.5973	C49H86O15	DGDG(18:2/16:1)	11.64 ± 1.29 ^0.010^ 7.85 ± 1.17	-	0.00 ± 0.00 4.54 ± 4.83	3.86 ± 1.41 ^0.028^ 4.22 ± 0.27
959.5925	C49H86O15	DGDG(18:1/16:2)	-	-	13.33 ± 3.62 4.44 ± 4.01	16.29 ± 2.90 13.36 ± 2.70
959.5973	C49H86O15	DGDG(18:0/16:3)	-	-	-	2.83 ± 0.14 ^0.001^ 2.62 ± 0.11
961.6123	C49H88O15	DGDG(18:2/16:0)	11.72 ± 0.82 12.32 ± 1.63	-	0.00 ± 0.00 0.81 ± 1.40	14.28 ± 1.06 ^0.003^ 12.66 ± 0.52
961.6123	C49H88O15	DGDG(18:1/16:1)	-	-	14.60 ± 4.28 ^0.006^ 17.73 ± 4.25	0.77 ± 0.07 0.57 ± 0.52
963.6238	C49H90O15	DGDG(18:1/16:0)	-	-	5.78 ± 1.96 ^<0.001^ 16.37 ± 1.69	-
975.5297	C51H78O15	DGDG(16:4/20:5)	-	5.32 ± 1.83 ^0.011^ 7.52 ± 1.28	-	-
977.5501	C51H80O15	DGDG(16:3/20:5)	-	3.98 ± 2.53 ^0.032^ 5.50 ± 0.57	-	-
979.5641	C51H82O15	DGDG(16:2/20:5)	-	39.44 ± 12.07 ^0.046^ 40.63 ± 5.74	-	-
981.5791	C51H84O15	DGDG(16:1/20:5)	-	30.46 ± 10.97 22.14 ± 1.51	-	-
983.5930	C51H86O15	DGDG(16:1/20:4)	-	1.73 ± 0.71 1.29 ± 0.18	-	-
981.5791	C51H84O15	DGDG(18:3/18:3)	30.56 ± 1.61 ^0.003^ 23.21 ± 1.68	-	7.30 ± 1.84 4.01 ± 1.93	0.00 ± 0.00 ^<0.001^ 0.95 ± 0.06
983.5966	C51H86O15	DGDG(18:3/18:2)	13.31 ± 1.18 13.90 ± 1.99	-	-	0.40 ± 0.37 ^0.002^ 1.24 ± 0.09
983.5978	C51H86O15	DGDG(18:1/18:4)	-	-	2.71 ± 0.87 ^0.046^ 2.34 ± 0.90	-
985.6111	C51H88O15	DGDG(18:2/18:2)	10.41 ± 0.31 12.84 ± 4.00	-	-	-
		Sum DGDG/∑GLs	20.11 ± 1.92 ^0.002^ 24.70 ± 7.29	16.47 ± 5.81 ^0.044^ 20.82 ± 2.62	11.62 ± 2.68 ^0.016^ 17.54 ± 7.05	25.37 ± 3.36 ^<0.001^ 36.79 ± 3.43
**SQDG**						
553.2611	C25H46O11S	SQMG(16:1/0:0)	-	N.Q.	-	-
583.3101	C27H52O11S	SQMG(18:0/0:0)	-	N.Q.	-	-
761.4517	C39H70O12S	SQDG(16:1/14:1)	-	0.00 ± 0.00 ^0.002^ 7.20 ± 2.09	N.Q.	-
789.4842	C41H74O12S	SQDG(16:1/16:1)	-	0.00 ± 0.00 ^<0.001^ 10.48 ± 1.20	-	-
791.4985	C41H76O12S	SQDG(16:0/16:1)	-	91.79 ± 20.27 73.87 ± 6.79	26.15 ± 3.03 ^0.037^ 23.80 ± 5.30	7.59 ± 0.37 ^0.040^ 8.24 ± 2.03
793.5102	C41H78O12S	SQDG(16:0/16:0)	-	-	2.52 ± 2.44 ^0.004^ 6.78 ± 0.99	17.15 ± 0.27 9.83 ± 2.69
813.4788	C43H74O12S	SQDG(18:4/16:0)^§^	-	0.85 ± 0.86 0.67 ± 0.58	-	-
815.4990	C43H76O12S	SQDG(18:3/16:0)	34.26 ± 4.14 32.94 ± 1.40	0.00 ± 0.00 0.09 ± 0.16	46.75 ± 5.76 ^0.013^ 38.87 ± 3.70	25.80 ± 2.11 ^<0.001^ 36.93 ± 1.73
817.5153	C43H78O12S	SQDG(18:2/16:0)	50.35 ± 3.98 ^0.020^ 53.63 ± 3.95	-	5.57 ± 9.65 11.67 ± 3.15	42.39 ± 1.17 ^0.002^ 39.33 ± 2.25
817.5153	C43H78O12S	SQDG(18:1/16:1)	-	-	0.72 ± 1.25 ^0.004^ 5.95 ± 1.72	-
819.5285	C43H80O12S	SQDG(18:1/16:0)	-	-	6.20 ± 0.59 ^<0.001^ 12.72 ± 1.63	6.14 ± 0.37 ^0.008^ 2.96 ± 0.57
819.5311	C43H80O12S	SQDG(18:0/16:1)	-	-	12.30 ± 8.57 ^0.034^ 0.00 ± 0.00	N.Q.
821.5436	C43H82O12S	SQDG(18:0/16:0)	14.99 ± 2.99 10.57 ± 2.95	-	-	1.05 ± 0.91 ^0.005^ 2.74 ± 0.37
835.4669	C45H72O12S	SQDG(18:4/16:3)	-	-	N.Q.	-
837.4828	C45H74O12S	SQDG(18:3/16:3)	-	1.77 ± 0.52 1.27 ± 0.15	N.Q.	-
839.4991	C45H76O12S	SQDG(16:0/20:5)	-	3.82 ± 1.67 ^0.021^ 5.22 ± 0.86	-	-
843.5279	C45H80O12S	SQDG(18:3/18:0)	-	0.00 ± 0.00 0.13 ± 0.23	-	-
859.4709	C47H72O12S	SQDG(18:3/20:4)	N.Q.	-	-	-
861.4869	C47H74O12S	SQDG(18:2/20:4)	N.Q.	1.81 ± 1.17 1.10 ± 0.07	-	-
		Sum SQDG/∑GLs	2.49 ± 0.45 ^0.004^ 3.59 ± 0.57	7.23 ± 1.77 ^0.035^ 8.95 ± 1.08	4.20 ± 1.31 ^0.007^ 4.17 ± 0.69	5.51 ± 0.29 ^0.003^ 7.62 ± 0.73
**PG**						
573.2842	C28H47O10P	PG(20:4)^§^	-	N.Q.	-	-
717.4701	C38H71O10P	PG(16:1/16:1)	-	24.10 ± 9.89 20.79 ± 4.46	-	-
719.4868	C38H73O10P	PG(16:0/16:1)	-	47.32 ± 17.43 50.79 ± 6.21	-	-
741.4736	C40H71O10P	PG(18:3/16:1)	59.30 ± 4.79 ^0.001^ 29.31 ± 1.53	-	10.03 ± 0.96 ^0.004^ 12.78 ± 1.75	0.00 ± 0.00 ^<0.001^ 1.06 ± 0.10
743.4875	C40H73O10P	PG(18:2/16:1)	21.09 ± 0.30 ^<0.001^ 51.61 ± 6.60	0.30 ± 0.52 1.39 ± 1.20	-	-
743.4865	C40H73O10P	PG(18:3/16:0)	-	-	33.89 ± 2.36 23.55 ± 3.48	18.59 ± 4.37 ^0.002^ 28.93 ± 1.21
745.5013	C40H75O10P	PG(18:2/16:0)	19.56 ± 1.27 18.96 ± 6.63	-	23.90 ± 6.37 13.80 ± 0.92	72.58 ± 3.81 61.80 ± 4.67
747.5157	C40H77O10P	PG(18:1/16:0)	-	-	23.57 ± 11.18 ^0.013^ 42.93 ± 9.84	4.30 ± 0.69 3.39 ± 2.37
773.5251	C42H79O10P	PG(18:1/18:1)	-	25.03 ± 16.82 23.53 ± 3.28	8.73 ± 1.18 6.93 ± 2.35	4.57 ± 1.73 4.40 ± 0.58
799.5103	C44H81O10P	PG(38:3)^§^	-	-	N.Q.	-
827.4863	C40H78O13P2	PGP(34:1)^§^	-	3.28 ± 1.71 3.53 ± 0.79	-	-
		Sum PG/∑GLs	0.69 ± 0.07 ^0.004^ 0.98 ± 0.20	1.19 ± 0.55 0.94 ± 0.15	1.10 ± 0.24 ^0.006^ 0.92 ± 0.17	1.86 ± 0.20 ^0.027^ 2.16 ± 0.19
**PC**						
800.5475	C42H78NO8P	PC(16:1/18:2)/(16:0/18:3)	13.57 ± 11.25 23.55 ± 15.14	-	-	-
802.5569	C42H80NO8P	PC(16:0/18:2)	N.Q.	-	-	-
820.5149	C44H74NO8P	PC(18:3/18:4)	-	-	N.Q.	-
824.5468	C44H78NO8P	PC(16:1/20:4)	21.01 ± 18.03 13.47 ± 3.87	99.95 ± 41.18 99.94 ± 7.01	44.40 ± 41.08 46.43 ± 53.07	-
826.5616	C44H80NO8P	PC(16:0/20:4)	65.20 ± 55.44 62.80 ± 19.15	-	-	-
826.5596	C44H80NO8P	PC(18:3/18:1)	-	-	55.52 ± 56.08 53.69 ± 63.25	-
888.4792	C50H70NO8P	PC(42:15)	-	N.Q.	-	-
**PE**						
686.4751	C37H70NO8P	PE(16:1/16:1)	-	54.38 ± 22.61 ^0.043^ 46.91 ± 11.13	-	-
688.4931	C37H72NO8P	PE(16:1/16:0)	92.89 ± 64.61 ^0.034^ 0.00 ± 0.00	43.27 ± 20.70 47.08 ± 12.81	-	-
714.5009	C39H73NO8P	PE(34:2)	6.87 ± 11.90 0.00 ± 0.00	2.13 ± 0.55 6.00 ± 2.01	-	-
736.4888	C41H71NO8P	PE(18:3/18:2)	-	-	31.53 ± 24.95 18.63 ± 19.18	-
736.4888	C41H71NO8P	PE(18:4/18:1)	-	-	1.10 ± 1.91 11.89 ± 13.07	-
738.5074	C41H74NO8P	PE(18:3/18:1)	-	-	43.77 ± 35.83 36.39 ± 35.30	-
740.5254	C41H76NO8P	PE(18:2/18:1)	-	-	23.55 ± 21.76 32.98 ± 29.30	-

**Table 3 ijms-22-11277-t003:** Unsaturation index (UI = [No. of double bonds in fatty acyls] × [GL content]/[Chla content]) calculated for the different glycerolipid classes and the sum of them for the four organisms studied here. Values are given for LL/HL light conditions. K, *Klebsormidium*; M, *Microcoleus*; H, *Haslea*.

Glycerolipid Class	*K. flaccidum*	*Oocystis* sp.	*M. vaginatus*	*H. spicula*
MGDG	14.12/10.47	12.87/17.88	11.90/15.05	32.43/32.09
DGDG	4.11/4.09	1.42/2.91	5.74/8.49	8.17/11.60
SQDG	0.19/0.23	0.21/0.33	0.45/0.71	0.73/1.14
PG	0.08/0.09	0.06/0.07	0.17/0.21	0.12/0.10
PC + PE	0.04/0.05	0.02/0.04	-/-	0.25/0.14
Total	18.53/14.92	14.59/21.28	18.26/24.46	41.69/45.06

**Table 4 ijms-22-11277-t004:** Band assignments for Raman spectra of three algae and one cyanobacterium. M, *Microcoleus vaginatus*; H, *Haslea spicula*; K, *Klebsormidium flaccidum*; O, *Oocystis* sp.

Wavenumber (cm^−1^)	Assignment	Organism	Relative Intensity	References
666.7	DNA ring breathing Phycobilins?	M	0.03	[31]
728.4	Tetrapyrrol ring Chls δ(CNH), β(NH), δ(ring)	H	0.06	[28,36]
815.2	Ring vibration from proteins δ(CNH), β(NH), δ(ring)	M	0.04	[28,31,36]
1004.2	Ring from Phe breathing C-H bending from Carotenoids	H, K, O, M	0.17	[28,31,33,40]
1157.5	ν(C-H) stretch from Carotenoids ν(C-C) from Cars	H, K, O, M	0.64, 0.57, 0.65, 1.00	[27,31,32,33]
1185.5	δ (C-H) cars ν(C-O) stretch saccharides	O, K, (H)	0.22, 0.21	[27,31,33]
1212.5	δ (C-H) cars/ν_as_(PO_2_) stretch ν (N-C) Chl a	O, K, (H)	0.11, 0.10	[27,34]
1283.8 1268.4, 1270.4	Amide III, (C-N) stretching plus (–N-H) bend/cis C=C bending in plane	M, K, O H	0.13, 0.05 0.07	[28,32,33,34]
1327.2	δ(C-H)/ν(C-N) from Chls	H, K, O, M	0.01, 0.09, 0.09, 0.09	[11,27,33]
1370.6	δ(C-H) proteins/porphyrin ring breathing	M	0.15	[11,36]
1445.8	δ(CH_2_) scissoring deformation (FAs, oleic, cars) ν(CO_2_) ν(C=C)?	H, K, O, M	0.07, 0.07, 0.07, 0.09	[27,28,29,31,32]
1524.9	ν(C=C) Carotenoids and fatty acyl chains of glycerolipids	H, K, O, M	1.00, 1.00, 1.00, 0.98	[27,28,31,32]
1582.8	Symmetric stretch of amide I?	M	0.18	[28,34]
1605.9	Amide I	H, K, O	0.06, 0.06, 0.06	[33]
1629.1	Amide I β-sheet/ ν(C=O) stretching from amide and carboxyl ν(C=C) stretch	M	0.36	[31,32,33,34,35]
1654.1 1681.1	Amide I β-sheet ν(C=C) from lipids	H K, O	0.04 0.02, 0.03	[32,33]
~2930	Carbohydrates (Galactose?)	H, K, O, M	0.78, 0.26, 0.66, 0.47	[41]

**Table 5 ijms-22-11277-t005:** Band assignments for FTIR spectra of three algae and one cyanobacterium. M, *Microcoleus vaginatus*; H, *Haslea spicula*; K, *Klebsormidium flaccidum*; O, *Oocystis* sp.

Wavenumber (cm^−1^)	Assignment	Organism	Relative Intensity	References
1031 1040 1045 1060	ν(C-O) Sterols Carbohydrates ν(Si-O)	M K O H	0.73 1.00 0.90 1.00	[5,40,42,43]
1143	ν(C-O-C) from carbohydrates and esters	H	0.67	[27,28]
1220	ν_as_(P=O) ν(C-O) from carbohydrates	H	0.56	[24,27,29,34,43]
1240	ν_as_(P=O) from nucleic acids and phospholipids	K, O, M	0.48, 0.58, 0.43	[24,27,34,43]
1395	δ_s_(CH_2_, CH_3_) of proteins and lipids ν_s_(C-O) of carboxylic groups	H, K, O, M	0.39, 0.44, 0.48, 0.43	[27,28,29,43,44]
1452	δ_as_(CH_2_, CH_3_) of proteins and lipids	H, K, O, M	0.39, 0.42, 0.48, 0.40	[27,28,29]
1537	ν(-CO) stretching of α-sheet amide II -N-H bend/ C-N stretching amide II	H, K, O, M	0.57, 0.70, 0.80, 0.81	[27,28,34,43]
1638	Amide I, β-sheet, ν(C=O) ν(C=C) stretching, band II	H, K, O, M	0.70, 0.87, 1.00, 1.00	[27,28,34]
1744	ν(C=O) stretching from FAs and esters	H	0.27	[5,27,29,34,43,44]
2856	ν_s_(C-H) from lipids (CH_2_, CH_3_)	H, K, O, M	0.29, 0.21, 0.25, 0.17	[27,29,43,44]
2927	ν_as_(C-H) from lipids, CH_2_	H, K, O, M	0.42, 0.32, 0.36, 0.28	[27,29,43]
3295	C-H stretching from carbohydrates (Galactose?) O-H water, N-H proteins	H, K, O, M		[43]

## Data Availability

Original data are available upon request from O.M.

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
