# Peer review of "Differential Membrane Lipid Profiles and Vibrational Spectra of Three Edaphic Algae and One Cyanobacterium"

_ijms, 2021, doi:10.3390/ijms222011277_

Round 1

Reviewer 1 Report

In this manuscript, the authors describe membrane lipid profiling by UPLC-MS of three algae and one cyanobacterium grown under two illumination intensities. The authors also applied FTIR and Raman approaches to understand insights into the molecular spectroscopy of diverse photosynthetic organisms. The manuscript is interesting and written well except for some minor errors. The authors need to be revised the manuscript following the comments below.

Comments:

Fig. 1 a, b, c are required to include scale bars. The scale bar used in Fig. 1d is vague so it also needs to revise.  

Supplementary figs S3 and S4 are not presented properly. The spectra in both Raman and FTIR need to be labelled to identify the results correctly.   

Author Response

We thank reviewer's suggestions as the manuscript is now clearly improved.

Reviewer comment 1: Fig. 1 a, b, c are required to include scale bars. The scale bar used in Fig. 1d is vague so it also needs to revise.

Answer: This figure has been redone. Scale bars are now included in all panels. The rough size (length) of the algal cells is now also provided in the figure legend (marked in red). However, the cell size of the Microcoleus vaginatus cells could not be determined due to their small size. 

Reviewer comment 2: Supplementary figs S3 and S4 are not presented properly. The spectra in both Raman and FTIR need to be labelled to identify the results correctly.  

Answer: The figures have been modified according to reviewer’s claim. They have also been enlarged for the sake of clarity.

Reviewer 2 Report

I want to congratulate the team that carried out this study.

figures 3 and 4 are a bit unclear, but cannot be modified because are an a lot of data - but please change the color of errors - to be as distinct as possible

Author Response

Reviewer comment 1: I want to congratulate the team that carried out this study.

Answer: We greatly thank reviewer’s positive valuation of our work, as well as his/her suggestions for improving the manuscript.

Reviewer comment 2: figures 3 and 4 are a bit unclear, but cannot be modified because are an a lot of data - but please change the color of errors - to be as distinct as possible

Answer: Colors of error bars have been changed. However, since we were unable to find a color combination which improved the resolution of the figures included in the main article, we have also included enlarged figures with a higher resolution, only panels a, b, c and d, in the supplementary information file as ANNEX A (at the end of the document). This issue is also indicated in the corresponding figure legends. We hope the accounted changes are satisfactory.